# Detection of Nitroaromatic Explosives in Air by Amino-Functionalized Carbon Nanotubes

**DOI:** 10.3390/nano12081278

**Published:** 2022-04-08

**Authors:** Claudio Ferrari, Giovanni Attolini, Matteo Bosi, Cesare Frigeri, Paola Frigeri, Enos Gombia, Laura Lazzarini, Francesca Rossi, Luca Seravalli, Giovanna Trevisi, Riccardo Lolli, Lucrezia Aversa, Roberto Verucchi, Nahida Musayeva, Muhammad Alizade, Sevinj Quluzade, Teimur Orujov, Francesco Sansone, Laura Baldini, Francesco Rispoli

**Affiliations:** 1IMEM-CNR Institute, Parco Area delle Scienze 37/A, 43124 Parma, Italy; giovanni.attolini@imem.cnr.it (G.A.); matteo.bosi@imem.cnr.it (M.B.); cesare.frigeri@imem.cnr.it (C.F.); paola.frigeri@imem.cnr.it (P.F.); enos.gombia@imem.cnr.it (E.G.); laura.lazzarini@imem.cnr.it (L.L.); francesca.rossi@imem.cnr.it (F.R.); luca.seravalli@imem.cnr.it (L.S.); giovanna.trevisi@imem.cnr.it (G.T.); rcrd.lolli@gmail.com (R.L.); 2Department of Physics and Earth Science, Ferrara University, Via Saragat 1, 44122 Ferrara, Italy; 3IMEM-CNR Institute, c/o FBK, Via alla Cascata 56/C, 38123 Povo-Trento, Italy; lucrezia.aversa@imem.cnr.it (L.A.); roberto.verucchi@imem.cnr.it (R.V.); 4Institute of Physics, Azerbaijan National Academy of Sciences, H. Javid Avenue 131, Baku AZ1143, Azerbaijan; nahida@mail.ru (N.M.); mehemmedelizade97@gmail.com (M.A.); sgseva92@gmail.com (S.Q.); timphysics1@gmail.com (T.O.); 5Department of Chemistry, Life Sciences and Environmental Sustainability, University of Parma, Parco Area delle Scienze, 17/a, 43124 Parma, Italy; francesco.sansone@unipr.it (F.S.); laura.baldini@unipr.it (L.B.); francesco.rispoli@unipr.it (F.R.)

**Keywords:** functionalized multiwalled carbon nanotubes (MWCNTs), characterization of functionalized carbon nanotubes, sensors for nitroaromatic explosives, TNT detection in air, resistive device

## Abstract

Nitroaromatic explosives are the most common explosives, and their detection is important to public security, human health, and environmental protection. In particular, the detection of solid explosives through directly revealing the presence of their vapors in air would be desirable for compact and portable devices. In this study, amino-functionalized carbon nanotubes were used to produce resistive sensors to detect nitroaromatic explosives by interaction with their vapors. Devices formed by carbon nanotube networks working at room temperature revealed trinitrotoluene, one of the most common nitroaromatic explosives, and di-nitrotoluene-saturated vapors, with reaction and recovery times of a few and tens of seconds, respectively. This type of resistive device is particularly simple and may be easily combined with low-power electronics for preparing portable devices.

## 1. Introduction

The development of chemical sensors and devices for trace detection of high explosives is an active area of research given the potential for this approach to increase antiterrorism actions and homeland security and to help foster secure methodologies for hidden landmines’ detection. 

Revealing the presence of vapors sublimated by solid substances at room temperature would be extremely valuable for simple and portable devices. Nitro-containing explosives, such as trinitrotoluene (TNT), pentaerythritol tetranitrate (PETN), and cyclotrimethylene trinitramine (RDX), are the most common class of explosives, especially for military use. Unfortunately, these substances are characterized by very low vapor pressure in the air at room temperature, such as 9 × 10^−9^ atm for TNT, 1.1 × 10^−11^ atm for PETN, and 5 × 10^−12^ atm for RDX [1].

Animals such as dogs and bees [2] are able to discover landmines, hidden explosives, or even explosive devices, but they need specific training and are often unable to be used for long research periods. Thus, detectors equipped with sensors that are able to reveal molecules of explosives in the air with a concentration of the order of a few parts per billion would be highly desirable.

Currently, several technologies can be applied for revealing explosives in trace amounts (see Moore [3] or Kolla [4]). Among these, mass spectrometry is widely considered the gold standard in chemical analysis [5,6,7] for detecting traces of explosives. Gas chromatography [8], infrared spectroscopy (IR) [9], cavity ring-down spectroscopy (CRDS), laser-induced breakdown spectroscopy (LIBS), surface-enhanced Raman scattering (SERS) [10] of explosive-contaminated surfaces and terahertz spectroscopy (THz) to penetrate objects, surfaces, and materials may all detect concealed explosives or weapons. However, most of these technologies require rather bulky instrumentation or time-consuming procedures and are not adaptable for portable devices for use in the field, or in dangerous environments.

Because of their unique electrical and optical properties, nanomaterials allow for the production of selective, sensitive, and simple sensors to detect explosives in traces. Due to their high surface to volume ratio, their sensitivity to chemical agents is usually very high [11]. The device size may be reduced to less than a few millimeters, thus permitting compact devices to work at low power. In this way, portable detectors may be used on unmanned air vehicles (UAVs) capable of exploring dangerous environments without direct human intervention [12,13].

Several sensors based on nanostructured metal-oxide or metal-sulfide particles have the capacity to detect gases in trace amounts. To increase the response and response-and-recovery speed, they typically work at high temperatures [14], but the detection of TNT vapor at room temperature was also reported [15]. The sensing mechanism is based on the vacancy defects adsorption of species such as oxygen, water, and organic molecules, leading to displacements of electrical carriers. Nevertheless, despite their high sensitivity, the absence of any functionalization results in poor sensor selectivity among the substances to be detected [15].

Carbon nanotubes (CNTs) are unique among nanoscale sensor platforms in their ability to detect the adsorption of as few as a single molecule of an analyte, as reported by Cognet, et al. [16]. CNTs may be covalently functionalized, becoming highly responsive to their physical and chemical environments. CNT-based chemiresistors displayed promising performance in explosive sensing [17]. To date, the majority of results in the literature demonstrated the capacity of the realized CNT-based sensors to reveal low concentrations of analyte molecules in a solution [18], but not directly in air, as highly desirable for preparing a compact device.

In this paper, we report the preparation and characterization of a multiwalled CNTs (MWCNTs)-based sensor with a simple structure for the detection of nitroaromatic explosives working at room temperature and in a gaseous phase.

## 2. Materials and Methods

### 2.1. MWCNT Preparation and Characterization

All reagents were commercially purchased from Merck KGaA (Darmstadt, Germany) and used without purification. IR spectra were recorded on a Spectrum Two FT-IR spectrometer (PerkinElmer Inc., Waltham, MA, USA). The spectra were collected from 64 accumulations in the wavenumbers range of 400–4000 cm^−1^ in transmission mode. Elemental analysis was performed using a FlashSmart apparatus (Thermo Fisher Scientific Inc., Waltham, MA, USA). After catalytic combustion at 900 °C in an oxygen atmosphere, the contents of carbon, nitrogen, and hydrogen were determined from CO_2_, NO_2,_ and H_2_O. The oxygen content was determined from what was left after oxidation. The results are reported as percent by weight of each element.

MWCNTs were synthesized by aerosol-assisted chemical vapor deposition (A-CVD) using heptane as the carbon source. The details of the growth process were reported by Musayeva et al. [19]. The MWCNTs’ purification from polycyclic aromatic compounds and amorphous carbon created during the growth process was carried out by keeping the MWCNTs for 20 min at 450 °C in air and then washing with xylene at 80 °C. 

MWCNTs 30–80 nm in diameter and several hundred micrometers in length were evidenced by Scanning Electron Microscopy analysis (SEM, Carl Zeiss Sigma HV, Oberkochen, Germany) (Figure 1a). Transmission electron microscopy (TEM JEOL220FS, Tokyo, Japan) observations, performed in STEM (scanning TEM) mode with a high-angle annular dark field detector, reported in Figure 1b, showed residual Fe nanoparticles, which were used as the catalyst for MWCNT growth and appear as white dots in the picture. The residual Fe nanoparticles were located inside the inner channel of the MWCNTs. The MWCNTs showed walls with a typical thickness of 20 nm [20]. 

### 2.2. Functionalization of MWCNTs 

To obtain MWCNTs capable of selectively interacting with TNT, we chose to functionalize their surfaces with short alkyl chains terminating with NH_2_ amino groups. The electron-rich amino group is well-known to interact strongly with an electron-poor aromatic ring of trinitrotoluene [21]. Following the procedure described by Wepasnick et al. [22], the first step of the MWCNT functionalization consisted of an oxidative acid treatment with a 3/1 H_2_SO_4_(95%)/HNO_3_(69%) mixture to introduce COOH carboxylic groups on their surface and obtain COOH−MWCNTs (Figure 2). The 5 h long reaction was performed under sonication in order to increase defects, where the oxidation process could take place [23]. 

For providing the MWCNTs’ surface with amine groups (NH_2_−C_2_−MWCNT in Figure 2), we selected ethylenediamine (EDA) as a reagent that could react at one of the two ends with the carboxylic groups present on the surface of the COOH−MWCNTs, forming an amide bond. The coupling was performed in the presence of N,N′−dicyclohexylcabodiimide (DCC) as a coupling reagent. As shown in Appendix A, the functionalization also improved the dispersibility behavior of MWCNTs, favoring the deposition process.

We also performed coupling between COOH−MWCNTs and hexamethylenediamine (HMDA) to investigate the effect of a longer distance between the MWCNT surface and NH_2_ active units on the interaction process with the analyte in the presence of DCC for 72 h (NH_2_−C_6_−MWCNT, see SI for the experimental details). However, we found that both NH_2_ groups of the longer HMDA chain could react with two different carboxylic functions present on the surface of either the same (Appendix A) or another MWCNT (see XPS analysis below). As a consequence, the availability of NH_2_ active units for the sensing activity was reduced.

## 3. Results and Discussion

### 3.1. Infrared (IR) Analysis

An infrared (IR) spectroscopic analysis of COOH−MWCNT and NH_2_−C_2_−MWCNT samples was carried out to confirm the functionalization process (Figure 3). The bands corresponding to C=C and C=O stretching vibrations for COOH−MWCNT appeared at 1636 and 1729 cm^−1^, respectively (Figure 3, middle curve). As a confirmation of the amide bond formation in the spectrum of NH_2_−C_2_−MWCNT (Figure 3, bottom curve), the carboxylic acid C=O stretching band at 1729 cm^−1^ was no longer present, and the stretching frequency of the amide C=O group was superimposed on the C=C stretching band at 1636 cm^−1^.

### 3.2. X-ray Photoelectron Spectroscopy 

X-ray photoelectron spectroscopy (XPS) was carried out in an ultra-high vacuum (UHV) system [24]. The details of the XPS analysis are reported in the Appendix A.

We analyzed different samples, namely pristine MWCNTs, COOH−MWCNT, and NH_2_−C_2_−MWCNT. Wide-range XPS spectra are shown in Appendix A. We found the presence of carbon (C1s core level), oxygen (O1s), and nitrogen (N1s), as shown in the wide-range spectra in Appendix A. The atomic percentages of these elements in the different materials are reported in Table 1. Oxygen was also present in pristine MWCNTs, as expected, due to adventitious C−O groups and water, but in oxidized CNTs its percentage was three times higher, suggesting the formation of other oxygen-based chemical groups. Nitrogen was present only in MWCNTs functionalized with EDA and HMDA.

C1s core level (see Figure 4, left) in all MWCNTs is characterized by a main peak related to C=C bonds located at about 284.5 eV, showing a typical slight asymmetry and a component related to C−C bonds and defects at about 285.5 eV [24]. Peaks located at 286.6 and 288.1 eV can be related to C−O/C−N and C=O/CON bonds.

A N1s core-level analysis of NH_2_−C_2_−MWCNT (Figure 4, right) showed two components located at 399.5 and 400.5eV due to NH_2_ amine and CON amide groups, respectively [25]. Their presence and weight, being about 50% each of the total N1s peak area, suggested a successful functionalization of MWCNTs, with each EDA chain linked to the MWCNT at one amine terminus (see schematic representation of NH_2_−C_2_−MWCNT in Figure 2). For NH_2_−C_6_−MWCNT, an amine/amide ratio of 20/80 was measured (Appendix A). This difference could be explained by considering the formation of additional amide units due to the reaction of both terminal HMDA amine groups with two different carboxylic groups, due to the longer chain of HMDA.

Table 1 summarizes the chemical composition in surface atomic percentages.

### 3.3. Sensor Structure Design and Test

Silver paste contacts were deposited on polyceramic glass substrates and dried. Successively amine-functionalized MWCNTs suspended in pure water at a concentration of 0.1 g/3 mL were drop-casted on glass substrates and successively dried at 50 °C for 20 min using a 1 cm^2^ device, as shown in Figure 5a. A few drops were sufficient to obtain a CNT network of 0.5 cm^2^ with resistance around 2 kΩ. The realized devices showed resistance values ranging from a few tens of kΩ to one kΩ, depending on the amount of MWCNTs deposited. The sensitivity, measured as a relative variation of resistivity under different gases, was independent of the sensor resistance.

The test of the sensor sensitivity on the saturated TNT and DNT vapors was carried out according to the scheme in Figure 6: a calibrated N_2_ gas flux at a rate of 1 cm^3^/s either flowed directly on the sensor surface or passed through a container with some grams of TNT or DNT. The ampoule, measuring chamber, and tubes carrying the N_2_ were all kept at room temperature, around 28 °C. The time to reach the gas saturation for a given volume was dependent on the temperature and TNT-exposed area. We roughly estimated that the condition of equilibrium between the vapor pressure and solid TNT could be reached in 10–20 min [26] considering the powdered state of TNT. The choice to use N_2_ was dictated by the need to avoid the effect of humidity in the air (estimated in a 0.2% H_2_O in weight), which was not stable with time. During the initial N_2_ flow, a decrease in sensor resistance of about 1% was observed due to the complete removal of humidity by N_2_ gas.

A Keithley 2635B source meter (Keithley Instruments, Cleveland, Ohio, United States) was used to measure the resistance change in the sensor after flushing the saturated TNT and gases. This was performed by measuring the voltage change across the sensor in series with a resistance R_C_ under a fixed-bias voltage of 5 V, as shown in Figure 5b. The same measurements were also performed by using the analog-to-digital converter input of an Arduino^®^ board, with a bias voltage of 5 V directly provided from the board, confirming the obtained results and the possible use on a portable device. 

Several tests were performed to guarantee that the measurements were reliable:The measuring chamber was kept in the dark and was shielded against electric noise. With or without N_2_ flux, the sensor signal was stable with noise of the order of 1 × 10^−4^.A sensor made with pristine MWCNTs that were tested with TNT and acetic acid did not show any change in resistivity.A test with a functionalized CNT network, performed by switching the gas flow through an empty ampoule, did not show any change in sensor resistivity.

As shown in Figure 7a, when the TNT gas was fluxed on the sensor, the resistance increased by 1% with a rise time in the order of 5 s. The recovery of the sensor resistance was reached after approximately 200 s, corresponding to the time to remove the gas in the TNT-containing ampoule. Several tests were repeated at time intervals of 20 min to permit the saturation in the bottle to recover, confirming the signal level and the rise and recovery times.

The sensor was subsequently tested with DNT vapor (Figure 7b). DNT has a vapor pressure at room temperature nearly two orders of magnitude higher than that of TNT [1], which should favor its detection. Despite this, the detector signal was lower (0.22% instead of 1% for TNT), probably due to lower electron deficiency in the DNT aromatic ring. These results suggested that the molecular mechanism responsible for sensing this type of molecule in the air involves the NH_2_ groups and originates from the electron deficiency of the analyte aromatic ring [27]. The same binding of electron-deficient TNT molecules to the electron-rich amine groups due to a charge-transfer Jackson–Meisenheimer (JM) was observed in TNT solution [28].

We also qualitatively tested the sensitivity of the obtained devices to several volatile substances, such as acetic acid, ammonia, water, and several common organic solvents (Figure 8). The concentrations in air of all these compounds were several orders of magnitude higher than those of the TNT and DNT vapors, so the experiments were carried out by rapidly exposing the sensor to a cotton swab saturated with the selected compound. Due to the expected acid–base interaction with NH_2_ functional groups, the sensor was particularly responsive to acetic acid. Changes in the sensor’s resistance also occurred with ethanol and methanol, presumably due to the formation of hydrogen bonds with the NH_2_ groups. The response to ammonia was likely due to the interaction between the analyte and the residual -COOH moieties present on the NH_2_−C_2_−MWCNTs. A confirmation of this hypothesis came from a sensor realized with a batch of NH_2_−C_2_−MWCNTs having a higher density of NH_2_ groups on their surfaces. In this case, we observed an increased response to acetic acid and a lower response to ammonia (Appendix A). All the curves showed similar rise and recovery times, defined as the time to reach 90% and 10% of the peak, respectively, of a few and tens of seconds, thus confirming the same sensing mechanism for the tested substances.

According to Zaporotskova et al. [29], the gas molecule adsorption on a single-walled CNT surface resulted in a change in CNT conductivity. The amino group acted as a charge transfer agent in CNTs, and the number of electrons transferred from the nanotubes to the NH_2_ molecule increased [30], thus decreasing the conductivity if the CNTs were an n-type semiconductor, or increasing it in the case of the p-type. In our study, all measurements showed an increase in the resistivity of the MWCNT network when exposed to several volatile substances. Due to the MWCNT wall thickness (>20 nm), we did not expect that such charge transfer in the proximity of the surface would substantially affect the MWCNTs’ conductivity. 

Because the MWCNTs were 100–200 μm long and quite tangled, as shown in Figure 1, the current in the path between the electrodes passes through many carbon nanotubes, and the device resistance was mostly dominated by the contact resistance between adjacent MWCNTs. Furthermore, due to the almost complete coverage of the MWCNT surface with functional groups (more than 3% atomic N), we expect that the capture of molecules at functionalized surfaces would increase the contact distance between adjacent MWCNTs, generating a resistance increase in the MWCNT network. A similar mechanism in MWCNTs was suggested by Zang et al. [31]. Another possible mechanism involves the formation of Schottky barriers at contacts between CNTs [29], assuming a metallic-type conductivity in part of the nanotubes.

A comparison with other sensors able to detect nitroaromatic explosives in the gas phase is shown in Table 2. The literature reports the capacity to detect saturated vapors of TNT and DNT, so the detection limit reported coincided with the concentration of saturated gas (≈9 ppb in air for TNT at 25 °C). Metal-oxide-based sensors seem promising for obtaining simple sensors, but normally they need to work at high temperatures (>200 °C) and the absence of functionalization limits their selectivity. Other techniques such as surface-enhanced fluorescence, florescence quenching, and surface plasmon resonance (SPR) are also promising but require more complex apparatuses than the resistive one proposed in this work.

## 4. Conclusions

Networks of MWCNTs with surfaces functionalized with NH_2_ amino groups were tested as sensors able to detect nitroaromatic explosives in a gas phase and then to detect the explosives directly in air. The obtained devices successfully revealed the saturated TNT and DNT vapors at room temperature (28 °C), with concentrations of 15 ppb and 1.5 ppm, respectively [1]. The reaction and recovery times were in the order of a few seconds and tens of seconds, respectively. Notably, the sensor’s response to TNT was five times higher than it was to DNT, despite the much higher DNT molecule concentration at room temperature. This was due to the higher affinity of a TNT molecule for NH_2_ amino groups. Considering the very good 0.1% signal stability under a pure N_2_ flux and the 1% resistance variation in the case of saturated TNT, we found that a sensitivity in the order of a few parts per billion in the case of TNT can be reached. For practical applications, these results obtained in a N_2_ atmosphere should be confirmed in controlled humidity and temperature conditions. Compared with other sensors based on surface-enhanced fluorescence and metal-oxide-based nanoparticles, amino-functionalized MWCNTs allow a resistive device working at room temperature with selectivity enhanced by functionalization to be easily combined with low-consumption electronics for preparing portable sensors.

## Figures and Tables

**Figure 1 nanomaterials-12-01278-f001:**
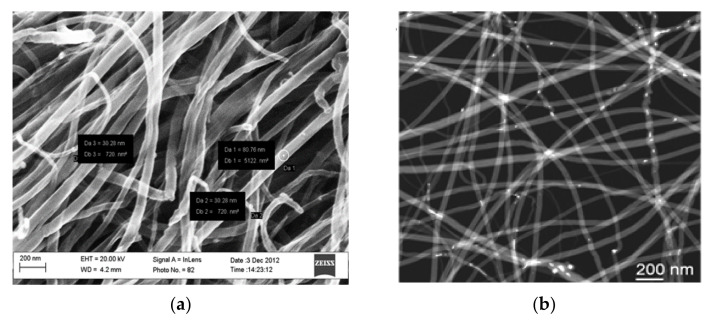
(**a**) Scanning electron microscope and (**b**) transmission electron microscope in scanning mode, 60,000× high-magnification images of purified MWCNTs.

**Figure 2 nanomaterials-12-01278-f002:**
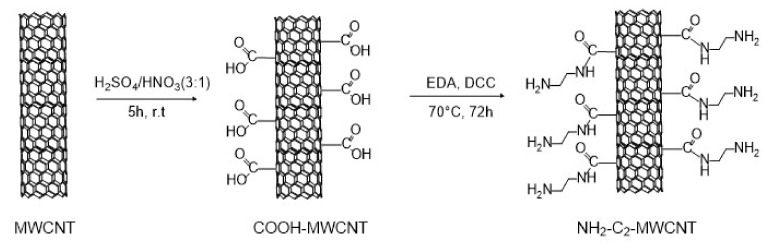
Functionalization of MWCNTs with amino groups.

**Figure 3 nanomaterials-12-01278-f003:**
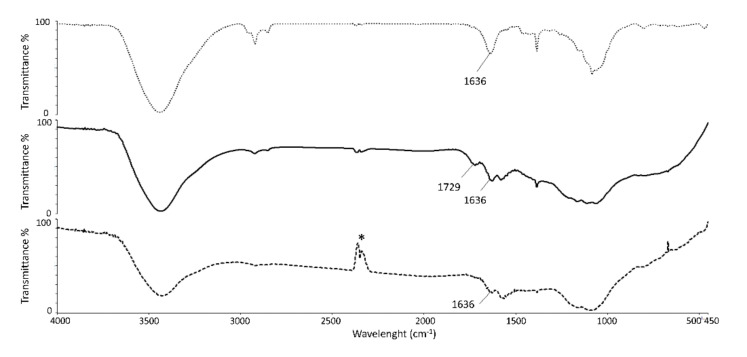
IR spectra of pristine MWCNTs (top curve), COOH−MWCNTs (middle curve), and NH_2_−C_2_−MWCNTs (bottom curve). * band of atmospheric CO_2_ not completely eliminated by background subtraction.

**Figure 4 nanomaterials-12-01278-f004:**
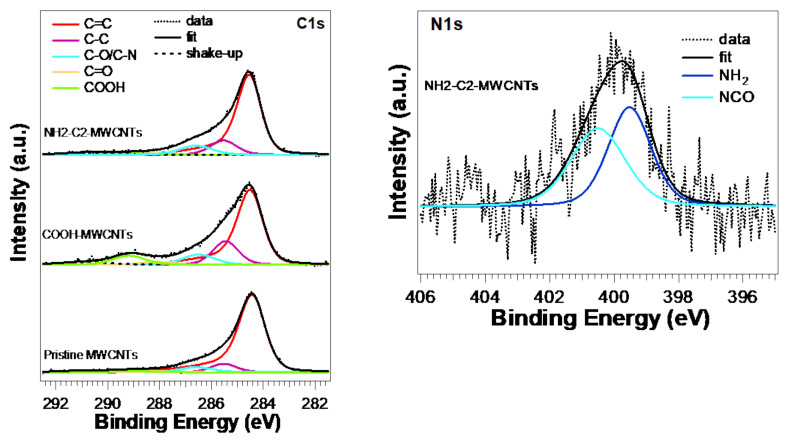
XPS core-level spectra and lineshape analysis of C1s (**left**) and N1s (**right**) of pristine MWCNTs, COOH−MWCNTs, and NH_2_−C_2_−MWCNTs.

**Figure 5 nanomaterials-12-01278-f005:**
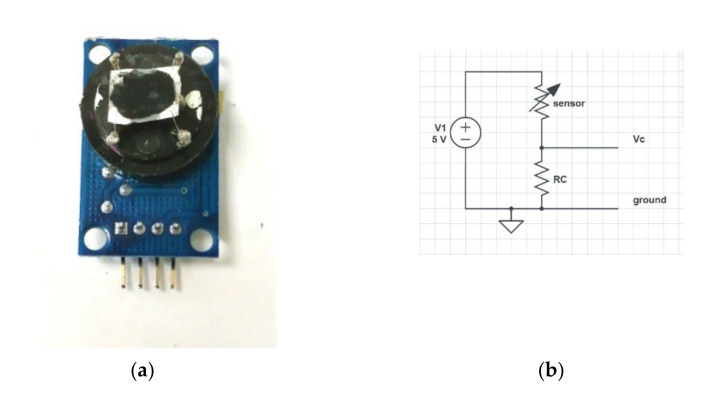
(**a**) MWCNT-based sensor and (**b**) the corresponding electrical circuit. The signal due to an increase in sensor resistance was detected on a pin Vc.

**Figure 6 nanomaterials-12-01278-f006:**
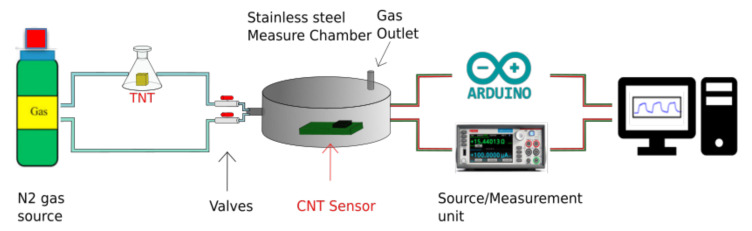
Sensor testing system.

**Figure 7 nanomaterials-12-01278-f007:**
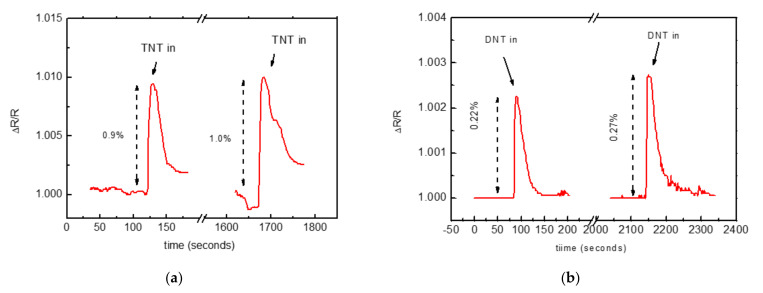
(**a**) Repeated changes in resistance of the NH_2_−C_2_−MWCNT based sensor under a rapid flux of TNT−saturated N_2_ gas. (**b**) Repeated changes in resistance of the NH_2_−C_2_−MWCN -based sensor under a rapid flux of DNT−saturated N_2_ gas. The sensor resistivity corresponded to 1.7 kΩ.

**Figure 8 nanomaterials-12-01278-f008:**
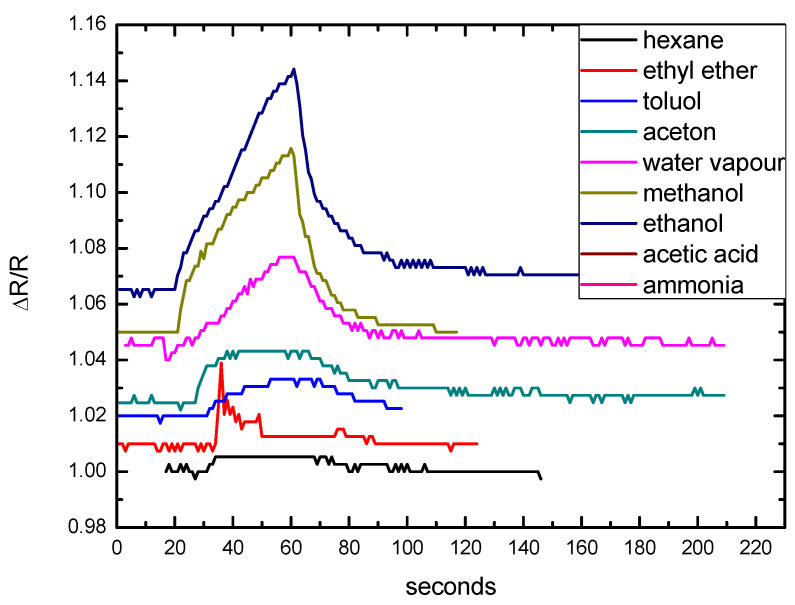
Test of sensitivity of the network of NH_2_−C_2_−MWCNT based sensor to different chemicals, obtained by exposing the sensor to a saturated cotton swab and showing similar rise and recovery times. The sensor base resistance was 820 Ohm.

**Table 1 nanomaterials-12-01278-t001:** Chemical composition in surface atomic percent for pristine MWCNTs, COOH−MWCNTs, and NH_2_−MWCNTs.

	% C	% O	% N
Pristine MWCNTs	94.0	6.0	-
COOH−MWCNTs	82.9	17.1	-
NH_2_−C_2_−MWCNTs	93.4	3.6	3.0
NH_2_−C_6_−MWCNTs	86.0	6.3	7.7

**Table 2 nanomaterials-12-01278-t002:** Summary of the recent literature and techniques able to detect nitroaromatic explosives in the gas phase.

Ligand	Sensing Device	Detection Limit	Reference
Amino-functionalized MWCNTs	Chemiresistive	Few ppb (T = 28 °C)	This work
Virus-phage litmus	Colorimetric	300 ppb	Cerruti et al. [32]
Carbazole-terminated black silicon	Surface-enhanced fluorescence	20 ppt (DNT)	Mironenko et al. [33]
WSe_2_/MoS_2_	Two-dimensional MoS_2_	80 ppb	Dhara et al. [34]
Poly(iptycenebutadiynylene)	Florescence quenching	5–7 ppb	Zhao [35]
Fe−ZnO	Chemoresistive	9 ppb	Marchisio et al. [36]
Core-shell ZnO/reduced graphene oxide (rGO)	Chemoresistive	1 ppb	Guo et al. [37]
Calix[4]arene-carbazole-containing polymers	Fluorescence-based	9 ppb	Barata. et al. [38]
TiO_2_ nanosheet	Chemiresistive gas sensor	9 ppb	Li et al. [39]
Alanine-based dansyl tagged copolymer	Fluorescence quenching	9 ppb	Kumar et al. [40]
Organic silane	Microcapacitive detection of adsorbed molecules	9 ppb	Gradišek et al. [41]
Silicon nanowires	Amino monolayer modified nanowire array	9 ppb	Liu et al. [42]
Polypyrrole-bromophenol blue	Quartz crystal microbalance	500 ppt	Eslami et al. [43]
Sulfonated dye-doped conducting polypyrrole	Chemiresistor	200 ppt	Ghoorchian et al. [44]
ZnO nanoparticles (NPs)	Chemiresistor	9 ppb	Ge et al. [45]

## Data Availability

Not applicable.

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
