# Peer review of "Detection of Nitroaromatic Explosives in Air by Amino-Functionalized Carbon Nanotubes"

_nanomaterials, 2022, doi:10.3390/nano12081278_

Round 1
Reviewer 1 Report
In this work, the authors prepared a resistive sensor to detect nitroaromatic explosives formed by amino-functionalized carbon nanotubes. However, the highlights of the article are not very clear. The material is not novel. The introduction part describes the background of gas sensor too much instead of the idea of this work. The innovation of the article does not meet the requirements of Nanomaterials submission. Therefore, it is not appropriate for publication in Nanomaterials.
1. The introduction is slightly incoherent too. For instance, paragraph 6 is about the disadvantage of some technologies while what the next paragraph write is nanomaterials instead of the technology used in this work. Besides, it is hard to say “Sensors based on nanostructured materials have the potential to satisfy all the requirements for an effective platform for the trace detection of explosives”.
2. This manuscript doesn’t read smooth, especially the part of gas sensor. The text and figure in this manuscript about the gas sensor need further polishing.
Author Response
Response to reviever #1
In this work, the authors prepared a resistive sensor to detect nitroaromatic explosives formed by amino-functionalized carbon nanotubes. However, the highlights of the article are not very clear.
It is not very clear what referee consider “highlights”. Is this the abstract? The key points of the paper are “resistive device”, that means very easy to implement and low consumption electronics, and gas detection at room temperature. We slightly modify the abstract to better evidence these points.
The material is not novel. The introduction part describes the background of gas sensor too much instead of the idea of this work. The innovation of the article does not meet the requirements of Nanomaterials submission. Therefore, it is not appropriate for publication in Nanomaterials.
We are a bit confused. There are thousand of papers dealing with explosive detection. A bit less for devices working in the gas phase, which is has the highest importance in view of his application on the field. To our knowledge this is the first time that a resistive device working at room temperature was able to detect TNT in the gas phase. The sensor is based on nanotechnologies, such as carbon nanotubes and proper surface functionalization. We cannot provide a clear response to the referee because we are not able to understand the criticism.
- The introduction is slightly incoherent too. For instance, paragraph 6 is about the disadvantage of some technologies while what the next paragraph write is nanomaterials instead of the technology used in this work. Besides, it is hard to say “Sensors based on nanostructured materials have the potential to satisfy all
the requirements for an effective platform for the trace detection of explosives”.
Most of the technologies able to reveal explosives in traces need rather bulky instrumentation or time
consuming procedures and are not adapt as portable devices. Nanostructured materials may provide very compact sensors thank to the material dimension, high sensitivity, the favourable surface/volume ratio. We tried to stress it again in the introduction. The phrase indicated by the referee has been removed, it is not necessary.
- This manuscript doesn’t read smooth, especially the part of gas sensor. The text and figure in this manuscript about the gas sensor need further polishing.
We provided a further English polishing and improved the figures, even it is not clear to which figures the comment refers.
Reviewer 2 Report
The manuscript submitted by Claudio Ferrari et al, reported the amino-functionalized carbon nanotubes are used to produce resistive sensors to detect nitroaromatic explosives by interaction with their vapors. The amino-functionalized MWCNT exhibits Sensitive response to several chemicals, including Acetic acid, TNT, DNT vapors, etc. Basically, I agree this article to be published, only after address minor revisions.
- The IR absorption spectra of pristine MECNT should be provided in Figure 3.
- The author claims that “It is worth to note that all the curves showed similar rise and the recovery times ….” In line 298, page 7. Yet from figure 8, it is obviously that there is huge difference between the rise and the recovery times between different chemicals, for instance, the recovery time of Acetic acid is about 150second while 10s for that of ethyl ether.
- Please verify how to calculate the detection limit of DNT and TNT vapors ( 1.8 and 9.0 ppm).
Author Response
Response to reviever #2
The manuscript submitted by Claudio Ferrari et al, reported the amino-functionalized carbon nanotubes are used to produce resistive sensors to detect nitroaromatic explosives by interaction with their vapors. The amino-functionalized MWCNT exhibits Sensitive response to several chemicals, including Acetic acid,
TNT, DNT vapors, etc. Basically, I agree this article to be published, only after address minor revisions.
- The IR absorption spectra of pristine MECNT should be provided in Figure 3.
The IR absorption spectrum of pristine MWCNT has been added to fig 3.
- The author claims that “It is worth to note that all the curves showed similar rise and the recovery times ….” In line 298, page 7. Yet from figure 8, it is obviously that there is huge difference between the rise and the recovery times between different chemicals, for instance, the recovery time of Acetic acid is about 150second while 10s for that of ethyl ether.
Wee agree with the referee comment that this statement is not clear. We need in fact to define how rise and recovery time are defined. We may define it as the rise time to reach 90% of the maximum level and the recovery time as the time needed to reach 10% of the maximum level. If we apply such definition we see that such times for the different substances are very similar or at least within a factor 2-3, suggesting that the same chemical mechanisms are responsible of the sensor resistance increase. We modified the sentence to specify the rise and recovery time definition.
- Please verify how to calculate the detection limit of DNT and TNT vapors ( 1.8 and 9.0 ppm).
We have not calculated the detection limit of DNT and TNT vapor, but in consideration of the very good 0.1% signal stability under a pure N2 flux and the 1% resistance variation in case of a saturated TNT vapor pressure, we concluded that a vapor pressure approximately 1/10 of the saturated gas could be detected, that is a few ppb of TNT in N2. We added a comment.
Reviewer 3 Report
The manuscript reports the obtaining of multi-walled carbon nanotubes functionalized with two types of amines and their investigation as resistive sensors for the detection of explosives vapors. The topic is very important and the manuscript is easy to read. Nonetheless some information is missing, the investigation seems a bit superficial and lacks quantitative evaluations with regards to the specificity. Revisions are necessary before recommending the manuscript for publication in Nanomaterials.
Specific comments
Section 3.3. What was the composition of the NH2-MWCNT solution deposited on glass substrates? Was it a pure aqueous dispersion, what concentration was it?
Figure 7 a graphs 7a and 7b seem to be identical, both show the response for TNT. Please check and correct.
Lines 222-223: the measurements were performed „at room temperature around 28 °C”? Typically is 20 C , what is the influence of the temperature on the response of the sensor? The temperature should be measured precisely.
The time for reaching the equilibrium was equilibrium between vapor pressure and solid TNT was „roughly 10-20 minutes” Later the authors mention that 9 ppb is the saturated concentration for TNT at 25 C? What is the concentration at 28º C, the temperature used in the experiments?
The authors should make it clear from the beginning if the target concentration to be detected is 9 ppb and if they think that the sensor is useful as yes/no response based on this amount or if it will be possible to make quantitative .
Lines244-245 What is the measurement units for the noise?
Figure 7: In order to compare it with Figure 8, the sensor resistance rather than the relative variation of the resistance should be illustrated.
Figure 8. Quantitative tests should be performed for specificity, in controlled humidity and temperature conditions.
Suggestion: for a discussion of the interaction of EDA-functionalized CNT with TNT , DNT and MNT: see also Sablok, K., Bhalla, V., Sharma, P., Kaushal, R., Chaudhary, S., & Suri, C. R. (2013). Amine functionalized graphene oxide/CNT nanocomposite for ultrasensitive electrochemical detection of trinitrotoluene. Journal of Hazardous Materials, 248-249, 322–328.
Author Response
Response to reviever #3
The manuscript reports the obtaining of multi-walled carbon nanotubes functionalized with two types of amines and their investigation as resistive sensors for the detection of explosives vapors. The topic is very important and the manuscript is easy to read. Nonetheless some information is missing, the investigation seems a bit superficial and lacks quantitative evaluations with regards to the specificity. Revisions are necessary before recommending the manuscript for publication in Nanomaterials.
Specific comments
Section 3.3. What was the composition of the solution deposited on glass substrates? Was it a pure aqueous dispersion, what concentration was it?
The NH2-MWCNTis not exactly a solution but a dispersion of MWCNTs suspended in pure water. We evaluate that in 3 ml we have 0.1g of functionalised MWCNTs. The functionalisation provides a good way of improving the dispersibility of the suspension, as visible in Fig. 1S.
We added some sentences.
Figure 7 a graphs 7a and 7b seem to be identical, both show the response for TNT. Please check and correct.
The referee is correct, we added the figure corresponding to DNT saturated vapor.
Lines 222-223: the measurements were performed „at room temperature around 28 °C”? Typically is 20 C , what is the influence of the temperature on the response of the sensor? The temperature should be measured precisely.
The measurements were performed at measured room temperature of 28 °C, it was quite hot. Usually authors vapor pressures at 25 °C, as reported by ref [1], considered as room temperature.
As a preliminary test of the dependence of sensitivity to temperature we heated the sensor to approximately 80 °C and we obtained an increase (20-30%) of the sensitivity with acetic acid. But the results were not systematically investigated, due to the difficulty of measuring the temperature on the sensor. We have calculated that an increase of approximately 50% of the vapor pressure of TNT at 28° with respect to 25 °C. This does not affect the conclusions of the paper about the ability to detect TNT at room temperature. We added some comments.
The time for reaching the equilibrium was equilibrium between vapor pressure and solid TNT was „roughly 10-20 minutes” Later the authors mention that 9 ppb is the saturated concentration for TNT at 25 C? What is the concentration at 28º C, the temperature used in the experiments?
According tp [1] the experimental values of the vapor pressure of bot TNT and DNT are determined with an error of 50% at T=25 °C, as determined by comparing and analysing experimental results.
Generally the vapor pressure is expressed by an exponential, see for instance J.C. Oxley, J.L. Smith, K. Shinde, J. Moran Propellants Explosives Pyrotechnics 30 (2005) 127–130.:
log10 (P/atm)=15.59-7145 /T(°K)
Which gives PTNT(T=25 °C)= 4.1x10-9 and PTNT(T=28 °C)= 7.1x10-9, which is a significant increase of the TNT vapor pressure. We modified the values in the conclusions, but in any case all the experimental values of TNT vapor pressure at room temperature are affected by a 50% error, as evaluated by [1].
The authors should make it clear from the beginning if the target concentration to be detected is 9 ppb and if they think that the sensor is useful as yes/no response based on this amount or if it will be possible to make quantitative .
The target concentration to be detected is at least the vapor pressure at room temperature, around 9 ppb, a result achieved in the present work. Sensors with a lower sensitivity will not be useful for the detection of TNT in the gas phase. In our case, due to the relatively noise free detector response, we evaluate that concentrations of a few ppb can be detected. We added a comment.
Lines244-245 What is the measurement units for the noise?
The evaluation is almost qualitative, based on the good stability of the base signal level, within 0.1% in a 10 minutes interval.
Figure 7: In order to compare it with Figure 8, the sensor resistance rather than the relative variation of the resistance should be illustrated.
We modified the figures according to the suggestion of the referee and reporting the sensor resistance in the captions.
Figure 8. Quantitative tests should be performed for specificity, in controlled humidity and temperature conditions.
We agree with the referee. We made qualitative measurements to test the sensitivity to several substances. Due to the method used they cannot be considered quantitative, because we expect a large difference in the gas concentrations. Nevertheless they are indicative of the different sensitivity with respect to the substances, that is a higher response corresponds to a higher sensitivity of the sensor.
We added a comment.
Suggestion: for a discussion of the interaction of EDA functionalized CNT with TNT , DNT and MNT: see also Sablok, K., Bhalla, V., Sharma, P., Kaushal, R., Chaudhary, S., & Suri, C. R. (2013). Amine functionalized graphene oxide/CNT nanocomposite for ultrasensitive electrochemical detection of trinitrotoluene.
Journal of Hazardous Materials, 248-249, 322–328. 
We expect that the same mechanism of the binding of electron-deficient trinitrotoluene (TNT) to the electron rich amine groups due to a charge-transfer Jackson–Meisenheimer (JM) is responsible of the resistivity increase in the MWCNT networks, as already reported in the paper of Sablok et al., which exhibited a sensitivity down to 0.1 ppb for TNT in solution. We added a comment and included the reference.
The discussion of the paper is very interesting. In the present case the resistivity increase of the sensor cannot be easily explained by the formation of TNT-ammine complexes. Other papers report the ability to detect in solutions even a few TNT molecules (Cognet et al. [14]). The detection in the gas phase appears more difficult, and the minimum detectable concentration in air is much larger.
Round 2
Reviewer 1 Report
This is NOT the first time that a resistive device working at room temperature was able to detect TNT in the gas phase. For your reference: Adv. Funct. Mater. 2016, DOI: 10.1002/adfm.201600592; J. Mater. Chem. 2011, 21, 7269; Adv. Mater. 2010, 22, 1900; ; IEEE Sens. J. 2013, 13, 1883. I am very sorry that I can not suggest accepting this manuscript.
Author Response
The statement that for the first time a resistive device working at room temperature is not in the manuscript but in the letter to the editor. We may agree that this statement may be misleading, since a few papers based on metal-oxide or ZnS nanostructures reported the capacity of revealing TNT in the gas phase but in general metal oxide based sensors require heating and the selectively is almost absent:
For instance:
In the Adv. Funct. Mater. 2016 paper “Contactless and Rapid Discrimination of Improvised Explosives Realized by Mn 2+ Doping Tailored ZnS Nanocrystals” the authors discuss the ability to discriminate explosives by comparing sensor based on different Mn doped ZnS nanocrystal. The sensing mechanism in is based on oxygen ion absorption as in many metal-oxide based sensor, without any selectivity. In fact we see in literature that the metal-oxide platform is not prevailing in the case of explosive sensors.
In our case we adopted the method of amino-functionalization that can permit some degree of selectivity.
In the Adv. Mater. 2010, 22, 1900 paper “2,4,6-Trinitrotoluene (TNT) Chemical Sensing Based on
Aligned Single-Walled Carbon Nanotubes and ZnO Nanowires” the authors really did not demonstrate the sensitivity to TNT in air at room temperature with concentration approximately 9 ppb, but to much higher TNT concentrations, obtained by heating the TNT. Moreover there is no selectivity, because there is no functionalization of the structures.
In the IEEE Sens. J. 2013 paper “Nitro-Aromatic Explosive Sensing Using GaN Nanowire-Titania Nanocluster Hybrids” the author claim the possibility of detecting TNT down to 500 ppt, apparently with a good selectivity, but again the interacting part of the device is based on TiO2 nanowires, that is a metal-oxide type sensor.
In conclusion we agree that some papers dealing with metal-oxide resistive sensor report the possibility of detecting TNT at room temperature (as already mentioned in the manuscript), but usually they show a poor selectivity because there is no functionalization. To try to answer to the referee doubts we added a comment in the conclusion “with selectivity enhanced by functionalization”.
In any case this confirms the originality of our paper.
Reviewer 3 Report
The issues raised in the review were adequately addressed, with the exception of the quantitative tests for specificity, in controlled humidity and temperature conditions. I suggest that the authors insert a phrase in conclusions acknowledging this limitation and the fact that future developments of the sensing device must include quantitative, more extensive specificity tests. .
Author Response
We agree with the suggestion of the referee and added a sentence in the conclusions. In fact the tests for TNT detection were made in N2 atmosphere.
“For practical applications these results obtained In N2 atmosphere must be replied in controlled humidity and temperature conditions”